# SwapTransformer: highway overtaking tactical planner model via imitation learning on OSHA dataset

## Abstract

This paper investigates the high-level decision-making problem in highway scenarios regarding lane changing and over-taking other slower vehicles. In particular, this paper aims to improve the Travel Assist feature for automatic overtaking and lane changes on highways. About 9 million samples including lane images and other dynamic objects are collected in simulation. This data; Overtaking on Simulated HighwAys (OSHA) dataset is released to tackle this challenge. To solve this problem, an architecture called SwapTransformer is designed and implemented as an imitation learning approach on the OSHA dataset. Moreover, auxiliary tasks such as future points and car distance network predictions are proposed to aid the model in better understanding the surrounding environment. The performance of the proposed solution is compared with a multi-layer perceptron (MLP) and multi-head self-attention networks as baselines in a simulation environment. We also demonstrate the performance of the model with and without auxiliary tasks. All models are evaluated based on different metrics such as time to finish each lap, number of overtakes, and speed difference with speed limit. The evaluation shows that the SwapTransformer model outperforms other models in different traffic densities in the inference phase.

## 1 Introduction

In the past decade, the field of autonomous driving has received lots of attention. Self-driving cars or autonomous vehicles (AV) represent a novelty of artificial intelligence (AI), robotics, computer vision, and sensor technology Grigorescu et al. (2020); Xiao et al. (2020); Zablocki et al. (2022). Many works focused on end-to-end learning approaches from camera to direct actions such as steering wheel, acceleration, and break Bojarski et al. (2016); Kim & Park (2017); Yang et al. (2018); however, there are many challenges such as lack of interpretability, data efficiency, safety and robustness, generalization, and trade-off between layers that make the end-to-end training less suitable for self-driving cars reliability. On the other hand, modular approaches break down the problem into different tasks such as perception and sensor cognition, motion prediction, high-level and low-level path planner, and motion controller Grigorescu et al. (2020); Atakishiyev et al. (2021); Teng et al. (2023). Self-driving is an intricate problem since it relies on tackling a set of different components such as following traffic laws, precise scene understanding, and performing safe, comfortable, and reliable maneuvers, such as lane changing, parking, etc.

In this study, we narrowed down and targeted the highway overtaking and lane-change problem. The ego vehicle is trying to match the velocity with the speed limit. The goal of overtaking on highways is to stay on the right-most lane unless there is a slow car in front of the ego vehicle. In that case, the ego can use the left lanes to pass other slow vehicles and return to the right-most lane. Figure 1 shows this overtaking task over two vehicles. Proper and on-time lane-change decision-making can help different aspects such as traffic flow, speed maintenance, reducing congestion, and avoiding tailgating Ulbrich & Maurer (2015); Nilsson & Sjöberg (2013). The current technology in commercial vehicles includes a Travel Assist controller that is explained in more detail in section 2.1. The Travel Assist controller is able to automatically move the car from one lane to another lane upon the driver's request. One of the key challenges for this controller is that it still needs the signaling request from the driver and is handled manually. The problem of lane-changing has already been

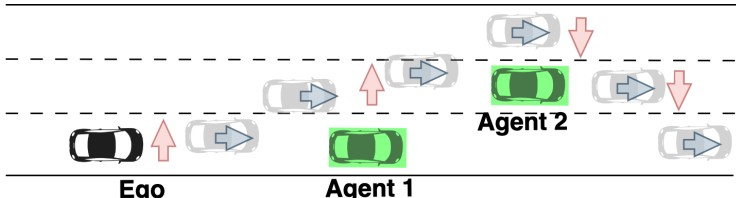

Figure 1: An example of an ego vehicle overtaking two agents by making left and right lane changes.

investigated by many approaches such as reinforcement learning Ronecker & Zhu (2019). However, usually, reinforcement learning techniques suffer from distributional shifts or gaps between the simulation environment and the real world Aradi (2020). Other approaches such as rule-based methods Lin et al. (2023); Scheidel et al. (2023) cannot fully handle edge cases in every scenario and situation. Moreover, they are computationally heavy to implement.

In this paper, we present a new AI approach to propose a lane change for the Travel Assist lane change module to perform overtaking and lane changing without human interaction. This AI module considers the safety of the ego based on training data. This paper considers environmental understanding by using a new swapping concept at its core. To have better generalizations, future points and car network predictions are proposed as auxiliary tasks. These auxiliary tasks are not involved in decision-making for the lane changes at the inference time; however, training these two subtasks improves performance at inference time. The model is tested successfully in a simulation environment.

The contributions of this study are as follows:

- OSHA Dataset: we are releasing a highway lane change dataset that is collected using the SimPilot simulation. This dataset is gathered based on the realistic behavior of vehicles in a simulation environment. About 9 million samples are collected as raw data and processed for the training aspect. OSHA dataset is available as a public resource AfterReview (2023).

- SwapTransformer: This model provides lane change predictions in highway scenarios. The swapping feature in the transformer is the key to better understanding the context and correlation between all agents including ego vehicle and other dynamic objects through features and time simultaneously.

- Auxiliary tasks: Our model solves secondary tasks such as future point and car network prediction sharing the same base with main tasks. We showed that while the output of auxiliary tasks does not explicitly affect the behavior of the ego vehicle, it helps during the training to better understand the dynamics of surrounding vehicles and road structure.

- Benchmarks: We made an excessive comparison for the proposed model and other baselines at inference time to evaluate metrics such as time-to-finish the scene, number of lane changes, and overtakes.

The rest of this paper is organized as follows. Section 1.1 discusses the related works in imitation learning. The Travel Assist controller, data collection, simulation environment, rule-based driver, and dataset are explained in sections 2.1 and 2.2. Sections 3.1 and 3.2 propose the idea of Swap-Transformer and how auxiliary tasks affect the training phase for better generalization and scene understanding. Evaluation is discussed in detail in section 4 by explaining simulation and training setups and assessing models. In the end, the conclusion is discussed in section 5.

## 1.1 RELATED WORKS

This section focuses on imitation learning approaches regarding lane-changing and overtaking scenarios and related technical approaches.

Imitation learning is one of the earliest and most successful methods for Autonomous Driving tasks. Authors in Zhang et al. (2023), Shin & Kim (2021), and Pashaki et al. (2022) have used different variations of Imitation Learning to solve overtaking or lane changing specifically.

Authors in Qasemabadi et al. (2023) showed that using a long short-term memory (LSTM) module for Lane Change (LC) prediction achieves a significant improvement in both adaptive cruise control (ACC) and cooperative adaptive cruise control. In another work Pashaki et al. (2022), the authors used a convolutional neural network (CNN) to show that this network is capable of learning the expert policy excellently.

In Zhang et al. (2023), the proposed approach solved discretionary lane changes considering different driving styles. They utilized CNNs to extract the ego and its surrounding vehicles' driving operational picture. Since their context movable objects are present, they also extract three traffic factors: speed gain, safety, and tolerance. These extracted features will be concatenated with the driving operational pictures and fed to an MLP. Spatio-temporal CNNs are used in Mersch et al. (2021), where the available data of eight vehicles (ego and seven surrounding vehicles in front and on both sides), stacked over time, is the input.

Authors in Wei et al. (2019) have developed lane-changing behavior detection using residual neural networks. Their proposed network solves this classification problem more accurately than the approaches combining support vector machines, principal component analysis, or vanilla CNN network, while trained end-to-end without a specific lane detection module using ResNetHe et al. (2015). They also point out that this data is in time sequences, so in the future, using modules like recurrent neural networks (RNN) or LSTMs will help analyze the entire data sequence. Furthermore, in Mozaffari et al. (2022), authors have classified the state-of-the-art deep learning approaches for vehicle behavior prediction, including lane change detection for ego or surrounding vehicles. This review further proves the usage of RNNs and CNNs in lane change behavior prediction.

With the extensive use of attention architecture in large language models and their proven capabilities in learning long-term dependencies and fusing features, authors in Zhu et al. (2023) and Du et al. (2023) used a combination of imitation learning and attention modules for autonomous driving tasks. In both of the aforementioned works the networks are inspired by Vision Transformers Dosovitskiy et al. (2021) since they use images from cameras to predict waypoints.

The popularity of transformer models Vaswani et al. (2017) shows why they are also an excellent choice for AD tasks Ngiam et al. (2021). Liang et al. (2022) uses an attention-based LSTM model on top of a C4.5 Quinlan (1993) decision tree to predict the lane change intention and execution. While in this work, the attention mechanism automatically extracts the critical information and learns the relationship between the hidden layers, in Ngiam et al. (2021), the trajectory prediction of other vehicles is solved by combining language models and trajectory prediction. Instead of predicting independent features for each agent, authors devised an approach employing attention to combine features across different dimensions (such as road elements, agent interactions, and time steps).

## 2  SYSTEM MODEL

### 2.1  TRAVEL-ASSIST CONTROLLER

In this study, we assumed that the ego vehicle is an electric vehicle equipped with the Travel Assist feature that includes three key components AfterReview (2021); Wolf & Burdick (2008): The first component is ACC which keeps and adjusts the speed based on what the driver set in the car and tries to consider traffic flow to avoid a collision with a vehicle in front. If a vehicle in front slows down, the ACC will slow down to keep a safe distance and once it is safe, it will accelerate to the speed that was set by the driver. The second component is lane-keeping assist (LKA) which uses cameras to detect the lane and keep the car in the lane. If the driver is not signaling, and the car drifts away from the center of the lane, LKA steers the vehicle and controls the car to the center of the lane. Finally, the last component is the lane change assist; once the driver triggers the signal, this module monitors the surrounding lanes for other vehicles. If it is safe, the car gradually moves to the requested lane by the driver; otherwise, it will reject the request.

Vehicles with Travel Assist utilize cameras and Radar. Specifically, Travel Assist can steer, accelerate, and decelerate the ego vehicle. The input for the lane change assist has three options, keep the current lane, change to the left lane, and change to the right lane. Figure 1 shows those triggers for the ego vehicle. Travel Assist controller is used by the rule-based algorithm for data collection. Moreover, the trained model interacts with the Travel Assist controller at the inference time

to control the ego vehicle. Travel Assist controller consists of seven different states 1) None, 2) Instantiated, 3) Ready-to-change 4) Start-movement 5) Interrupted 6) Success, and 7) Failed. The controller receives the lane change command in the None state and follows other corresponding states. More information about the Travel Assist controller is available in Section A.1

## 2.2 DATA COLLECTION

This section discusses the importance of data generation in regards to imitation learning used in Section 3. First, the simulation environment SimPilot is introduced, the rule-based algorithm is described, and lastly, the features of raw and processed data are detailed.

### 2.2.1 SIMPILOT: SIMULATION ENVIRONMENT

SimPilot is the name of the driving simulation which is built in-house based on the Unity engine Haas (2014). Unity provides the required graphics and physics for the ego vehicle and other dynamic and static objects. Sumo controls the traffic behavior in SimPilot in the background Behrisch et al. (2011). The SimPilot platform provides different cities such as Berlin and San Francisco, and training and evaluation scenes for highways. In this paper, highway scenes are used. SimPilot provides different types of sensors and observations based on real-world hardware and implementation. SimPilot is able to communicate with Python through a software called MLAgent Juliani et al. (2018). Using MLAgent, our AI model is able to control the Travel Assist module and hence control the ego vehicle. Specifically in this approach, observations are received through MLAgent and fed to the model, and then the output of the model is sent back to the SimPilot to control the ego vehicle.

### 2.2.2 RULE-BASED DRIVER

We developed a simple rule-based driver to control the ego for data collection purposes. Despite the fact that Sumo is capable of controlling ego, we opted to develop our custom ego driver. The primary motivation for developing this rule-based driver was to retrieve the lane change commands, as Sumo could not. In our approach, we considered the six vehicles surrounding the ego to decide on a lane change. These six vehicles are the vehicles in front and behind the ego in the current lane and the adjacent lanes if they exist. The state of each vehicle can be represented as $(x, y, v, \text{ID})$, where $(x, y)$ is the relative position to ego, $V$ is velocity, and ID is the lane ID. A lane change will happen only if all four conditions of *Safety*, *Speed gain*, *Availability*, and *Controller acceptance* are met (definitions can be found in A.3).

## 2.3 OSHA DATASET: RAW AND PROCESSED DATA

OSHA dataset is created by a combination of rule-based driver and SimPilot simulation for training and validation purposes. In this study, we collected raw data by utilizing the rule-based driver as an expert driver to collect sensor and lane information. To match the data requirement with vehicle hardware and sensors, only lane ID segmentation is collected as vision data. Other information is collected as ego vehicle data and object list. Different traffic densities vary from 5 to 35 vehicles per kilometer are used for data collection. The environment is seeded and after each episode, the ego car is located in a random lane and location on the scene. Three types of speed behavior as *slow*, *normal*, and *fast* are considered for other dynamic objects. The scene is partitioned into different segments with different speed limits. Data samples are explained in detail in Appendix A.4.

ego vehicle data is a tuple of $\langle v, s, L_{ID}, l, r, c \rangle$ where $v \in \mathbb{R}^+ \cup \{0\}$ is the ego velocity. $s$ is the speed limit associated with that part of the road that ego currently is driving with pre-defined values as integers in $[\frac{m}{s}]$ unit. $L_{ID}$ is an integer value for the current lane ID of ego. $l, r \in \{0, 1\}$ are boolean values indicating if left or right lanes are available, respectively. ego location $(x, y)$ are also collected in meters defined in a global coordinate system. ego location is used for the auxiliary task of the future position prediction. Also, all rule-based lane-change commands $(c)$, for the current, left, and right lanes are collected at each time step.

On the other hand, simulation provides information about other vehicles as well. This information for each vehicle is a tuple of $\langle v_k, x_k, y_k, L_{ID_k}, m_k \rangle$ where $v_k \in \mathbb{R}^+ \cup \{0\}$ is the velocity for vehicle $k$. $x_k, y_k \in \mathbb{R}$ are the positions for vehicle $k$ defined in a local coordinate where the ego is at the origin $(0, 0)$. The length of $k^{th}$ vehicle is defined as $m_k$ where $m_k \in \mathbb{R}, \forall k \in [1, 20]$. During the

Table 1: Dataset features for raw and processed data

| Description | Raw Data | Processed data |
|---|---|---|
| Number of pickle files | 900 | 1 |
| Pickle file size (single) | 34.1 MB | 61 GB |
| Images size | 5.7 MB (episode) | 35 GB |
| Total number of samples | 8,970,692 | 8,206,442 |
| Lane change commands | 5,147 | 69,119 |
| Left lane commands | 2,648 | 35,859 |
| Right lane commands | 2,499 | 33,260 |
| Transition commands | 0 | 1,468,115 |
| Number of episodes | 900 | 834 |
| Samples per episode | 10,000 | 9,839 (Average) |
| Speed limit values | $\{30, 40, \ldots, 80\}(\frac{km}{h})$ | $\{30, 40, \ldots, 80\}(\frac{km}{h})$ |
| ego speed range | $[0, 79.92](\frac{km}{h})$ | $[0, 79.92](\frac{km}{h})$ |

processing phase, the future positions are computed for each time step. Since SwapTransformer in section 3 requires future positions, speed, and lane change commands at each time step; 2.5 seconds (5 points each 500ms apart) of the future data from the recorded samples are extracted and linked to the corresponding timestep.

Since the number of lane changes to left and right lanes is noticeably smaller than the number of commands to stay in the current lane commands, the dataset becomes imbalanced. Before pre-processing the raw data, on average 20 lane changes occur in each episode of collected data with 20,000 steps. Hence, another pre-processing is performed to artificially add augmented lane changes to the original data. A new class called *Transition* is defined to highlight the lane-changing phase between two lanes and it reduces the effect of the imbalanced classes. In addition to the new class, the dataset is augmented to include more lane change commands. Since it takes about 20 steps for the controller to process the lane change command and start the maneuver, those steps are augmented as artificial lane changes.

Table 1 shows some details about the dataset used in this paper. More specifically, we show some differences between raw and processed data regarding the number of lane changes, the number of samples, and episodes. Speed limit and ego speed values are common between raw and processed data. The dataset is available here AfterReview (2023) with more details and features. More information about the OSHA dataset is available in Appendix A.4.

## 3 PROPOSED APPROACH

Our model utilizes two inputs; 1) Lane ID segmentation images, which identify and segment road lanes and demonstrate road curvature. 2) Object lists containing detailed information about the ego vehicle and surrounding cars. For the ego vehicle, this includes current velocity, lane ID, lane change status, lane availability, and speed limit. For other cars, it comprises velocity, relative position to the ego vehicle (x, y), lane ID, and vehicle length. These inputs are stacked in sequences of data, with a history length of five seconds, enabling the model to learn spatio-temporal patterns.

Next, lane ID images are encoded using a CNN encoder network (ResNet 18 He et al. (2016)), and object list information is processed through an attention-based module. This module employs self-attention across features and time, correlating them to capture fine-grained details and meaningful connections in the object list data.

The model also employs auxiliary tasks (dashed lines in Figure 2) to improve feature representation and gauge performance. One auxiliary task predicts the ego vehicle's future positions, while another, based on the attention module, generates a car graph representing vehicle distances using edges and nodes of a complete graph.

In the main tasks (solid lines in Figure 2), we use the output from the attention module. These outputs are then directed into two distinct MLPs: one is responsible for predicting future lane changes, while the other predicts future velocity. Future predictions serve to enhance our understanding of upcoming dynamic states. Specifically, we predict five future lane changes and velocities; however,

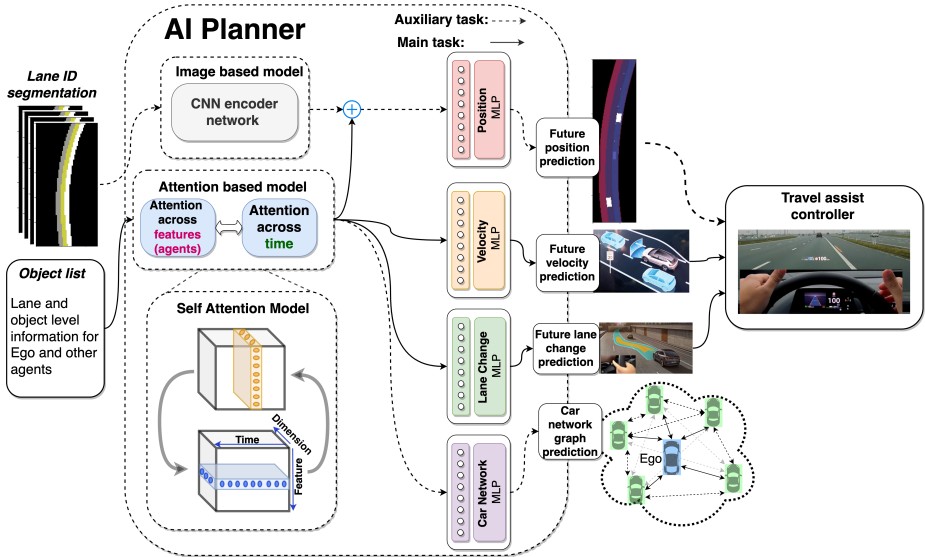

Figure 2: SwapTransformer architecture to interact with Travel Assist controller.

the Travel Assist controller only takes the first point to influence the vehicle's behavior Hu et al. (2022).

$$\mathcal{L}_{\text{CE, LC}} = -\frac{1}{N} \sum_{i=1}^{N} \boldsymbol{p}_i \log(\hat{\boldsymbol{f}}_i) \tag{1}$$

As shown in Equation (1), $\mathcal{L}_{\text{CE, LC}}$ assesses the model's ability to predict lane changes, including right lane change, left lane change, maintaining the same lane, and transitioning. It penalizes discrepancies between predicted probabilities ($\boldsymbol{p}_i$) and actual outcomes ($\boldsymbol{f}_i$) using cross-entropy loss.

$$\mathcal{L}_{\text{MSE, V}} = \frac{1}{N} \sum_{i=1}^{N} (\boldsymbol{v}_i - \hat{\boldsymbol{v}}_i)^2 \tag{2}$$

In addition to the lane change prediction, (2) assesses the model's ability to predict future vehicle velocities. It quantifies the accuracy of velocity predictions by comparing predicted velocities ($\hat{\boldsymbol{v}}_i$) to actual velocities ($\boldsymbol{v}_i$) for each sample. This loss is defined based on the mean squared error (MSE).

## 3.1 SwapTransformer

The SwapTransformer shown in Figure 3 introduces a novel approach to the well-known concept of self-attention Vaswani et al. (2017). Initially, we take input data, in the form of an object list, and process them through an embedding layer while also incorporating positional encoding. These inputs consist of a batch of data points where each data point is represented as a 3D tensor comprising time information, an arbitrary embedded dimension, and features sourced from the object list.

Once we have embedded and applied positional encoding to the input data, we employ two distinct types of transformer encoders with unique input dimensions. One encoder aligns its dimensions with the features present in the data, whereas the other matches the sequence length. Subsequently, the inputs are fed into the first encoder, and the output is then transposed. Subsequently, this transposed output is passed into the other transformer encoder. The same process will repeat for each encoder.

## 3.2 Auxiliary tasks

Solving auxiliary tasks in deep learning is a valuable strategy that offers several advantages. First, it acts as a form of regularization, preventing overfitting and enhancing model generalization. Second, it promotes feature learning, allowing the model to extract meaningful representations from the data. Addressing auxiliary tasks is not a novel approach in the realm of deep learning; it has been employed in various subdomains He et al. (2022); Chen et al. (2019); Mehta et al. (2018).

Figure 3: Time and feature swapping in SwapTransformer.

### 3.2.1 FUTURE POSE PREDICTION

We propose two auxiliary tasks, the first one is predicting the future positions of the ego vehicle, which aids in comprehending future dynamics. Our model predicts the $(C_x, C_y)$ coordinates of the control points of a Bezier curve, followed by fitting the Bezier curve. Bezier curves are commonly used in trajectory planning for efficient vehicle motion modeling and prediction. Their versatility and controllability make them a preferred choice for representing and generating smooth paths for autonomous vehicles Choi et al. (2008). By leveraging Bezier curves, our auxiliary task improves future pose predictions.

$$B(t) = (1-t)^4 \cdot C_0 + 4(1-t)^3 \cdot t \cdot C_1 + 6(1-t)^2 \cdot t^2 \cdot C_2 + 4(1-t) \cdot t^3 \cdot C_3 + t^4 \cdot C_4, \quad 0 \le t \le 1 \quad (3)$$

In the context of the Bezier curve (3), the parameter $t$ represents the parametric value or parameterization along the curve that ranges from 0 to 1. Typically, $t$ is used to interpolate between control points and generate the smooth path described by the Bezier curve. In this case, we omit the $C_0$ point because it starts at $(0, 0)$, and our model focuses on predicting the subsequent control points ($C_x$, $C_y$) that define the curve's shape and trajectory. We selected the quartic Bezier curve because of its ability to capture more complex and subtle curve shapes compared to lower-order Bezier curves, such as quadratic or cubic Bezier curves. We evaluated lower-order Bezier curves and the quartic curves yielded the lowest error compared to the ground truth curves. The loss function used for future pose prediction as the auxiliary task is:

$$\mathcal{L}_{BZ}(\boldsymbol{p}, \hat{\boldsymbol{p}}) = \frac{1}{N} \sum_{i=1}^{N} (||\boldsymbol{p}_i - \hat{\boldsymbol{p}}_i||_2^2 \cdot \boldsymbol{w}_i), \quad (4)$$

where $\mathcal{L}_{BZ}$ is the weighted loss value between the prediction $\hat{\boldsymbol{p}}_i$ and ground truth $\boldsymbol{p}_i$ for a batch with $n$ samples. $\boldsymbol{w}_i = e^{\frac{1}{K-i}} - 1$ is the weighted vector applied to different points of the predictions to have better results on farther points. $K$ represents an arbitrary constant value for the number of points in the line.

### 3.2.2 CAR NETWORK PREDICTION

Our secondary auxiliary task aids in comprehending the spatial relations and awareness among all vehicles within a given scene. We generate distance matrices that include all vehicles, including the ego vehicle as:

$$\boldsymbol{D} = \begin{bmatrix} 0 & d_{0,1} & d_{0,2} & \cdots & d_{0,19} & d_{0,\text{ego}} \\ d_{1,0} & 0 & d_{1,2} & \cdots & d_{1,29} & d_{1,\text{ego}} \\ d_{2,0} & d_{2,1} & 0 & \cdots & d_{2,19} & d_{2,\text{ego}} \\ \vdots & \vdots & \vdots & \ddots & \vdots & \vdots \\ d_{19,0} & d_{19,1} & d_{19,2} & \cdots & 0 & d_{19,\text{ego}} \\ d_{\text{ego},0} & d_{\text{ego},1} & d_{\text{ego},2} & \cdots & d_{\text{ego},19} & 0 \end{bmatrix}, \quad (5)$$

where $d_{i,j} \in \mathbb{R}^+ \cup \{0\}$ is the Euclidean distance between any pair of vehicles $i$ and $j$, including the ego and other objects. We should also mention that we mask out ($\boldsymbol{m}_{\text{vehicle}}$) distances when no vehicles are present in the scene to account for empty scenarios in the loss function. In this loss (6), $\boldsymbol{D} \in \mathbb{R}^{N \times N}$ represents the ground truth (actual) distances between vehicles, and $\hat{\boldsymbol{D}}$ represents the

distances predicted by our model.

$$\mathcal{L}_{\text{MSE, CN}} = \frac{1}{N} \sum_{i=1}^{N} \left( (\boldsymbol{D}_i - \hat{\boldsymbol{D}}_i)^2 \cdot \boldsymbol{m}_{\text{vehicle}} \right) \tag{6}$$

To ensure our model effectively learns all relevant features, we must back-propagate through this total loss (7). The components of this total loss include cross-entropy loss for various lane change classes, regression on velocity differences, regression on losses related to Bezier curves, and finally, regression on the Car Network loss.

$$\mathcal{L}_{\text{total}} = \mathcal{L}_{\text{CE, LC}} + \mathcal{L}_{\text{MSE, V}} + \mathcal{L}_{\text{BZ}} + \mathcal{L}_{\text{MSE, CN}} \tag{7}$$

## 4 EVALUATION

### 4.1 SIMULATION SETUP

After completing data collection and training of the SwapTransformer, we conducted an evaluation in a simulated environment to compare the set of models. During this evaluation phase, each model undergoes inference for 50 episodes in an unseen environment for a maximum of 20,000 steps (equivalent to 400 seconds) or until completing one lap around the track, whichever occurred first. Additionally, our assessment encompassed three distinct traffic scenarios, categorized as follows: low, medium, and high density with 5, 15, and 25 vehicles per km on the road, respectively.

### 4.2 TRAINING SETUP

Training is performed on a local cloud instance with a learning rate $\alpha = 0.0001$, batch size of 256 on 8 GPUs (Nvidia A100 - 80GB memory) in a distributed data-parallel manner. Adam Kingma & Ba (2014) is used as an optimizer during training alongside ResNet18 as the feature extractor for lane ID segmentation. All five models in table 2 were trained for 300 epochs taking approximately two days to train each model. More details about the training are available in Section A.6. The purpose of training five models with the selected features is to compare them and see the effect of each module on different performance metrics in the next section.

### 4.3 INFERENCE AND RESULTS

The first column of table 2 reports how each trained model is behaving at velocity prediction during inference time. A model with better speed following with respect to the speed limit has a lower metric. In the speed difference column of table 2, our model outperforms other models in low and medium traffic densities. Based on the content of table 2 SwapTransformer model outperforms all the other models in all traffic densities by finishing the loop fastest in all traffic scenarios.

The last column of table 2 shows the left overtake ratio in a highway environment. The average number of vehicles overtaken by each model is maximum with the SwapTransformer model in medium- and high-density traffic scenarios. It is noted that, in medium-traffic densities, all models exhibit a higher frequency of overtaking maneuvers as opposed to high-traffic densities. This disparity can be rationalized by the congestion in high-density traffic, which leaves limited opportunities for overtaking. Subsequently, after conducting comparisons among various models during the inference phase, we proceeded to execute the SwapTransformer model with auxiliary tasks to demonstrate its interaction with the SimPilot environment, as illustrated in Figure 4. The combination of auxiliary tasks

Table 2: Comparison between SwapTransformer (**ours**) and other baselines on different metrics.

| Metrics | 1) Speed difference (m/s) ↓ | | | 2) Time to finish (s) ↓ | | | 3) Left overtake ratio ↑ | | |
|---|---|---|---|---|---|---|---|---|---|
| **Traffic** | Low | Med | High | Low | Med | High | Low | Med | High |
| MLP (Baseline) | $3.19 \pm 0.7$ | $4.16 \pm 0.98$ | $4.37 \pm 0.77$ | $181.24 \pm 9.2$ | $192.02 \pm 7.78$ | $193.59 \pm 7.39$ | 0.04 | 0.02 | 0.06 |
| Transformer only | $2.11 \pm 0.46$ | $2.81 \pm 1.28$ | $\mathbf{3.45 \pm 1.66}$ | $170.27 \pm 4.23$ | $174.07 \pm 3.45$ | $179.69 \pm 7.18$ | 0.12 | 0.28 | 0.24 |
| Transformer + Aux. | $2.25 \pm 0.35$ | $2.9 \pm 0.75$ | $4.19 \pm 1.44$ | $171.84 \pm 3.36$ | $178.8 \pm 8.08$ | $193.05 \pm 11.78$ | **0.2** | 0.16 | 0.12 |
| Transformer + Swap | $2.73 \pm 0.99$ | $3.26 \pm 0.94$ | $4.2 \pm 0.98$ | $174.34 \pm 5.82$ | $182.32 \pm 7.85$ | $195.11 \pm 12.3$ | 0.1 | 0.18 | 0.1 |
| **Transformer + Swap + Aux. (Ours)** | $\mathbf{2.09 \pm 0.5}$ | $\mathbf{2.45 \pm 1.2}$ | $3.55 \pm 1.97$ | $\mathbf{168.7 \pm 1.99}$ | $\mathbf{169.97 \pm 2.47}$ | $\mathbf{176.93 \pm 9.32}$ | 0.16 | **0.38** | **0.3** |

and dimension-swapping exhibits synergistic effects, enhancing the model's performance by leveraging both strategies concurrently. However, when individually trained—either with Transformer + Auxiliary or Transformer + Swap—both models exhibit a tendency to underfit within the 300-epoch computational budget. In the swap transformer, we 'swap' dimensions between the time horizon and features. If trained longer, Transformer+Aux or Transformer+Swap by themselves would outpace the Transformer in evaluation. Hence, the Swap mechanism and auxiliary tasks are complementary in terms of faster convergence during training because of the additional signals they provide for each other Ruder (2017).

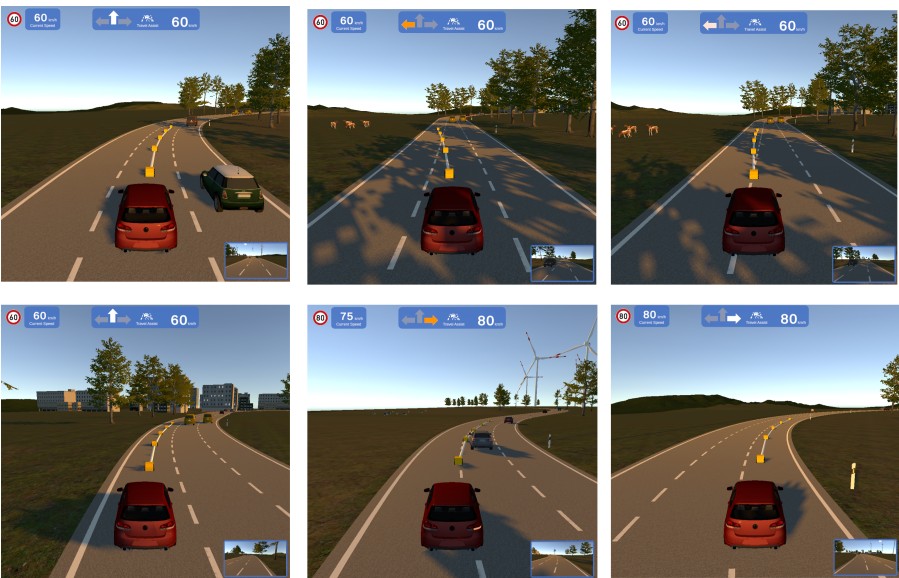

Figure 4: Frames of decision-making with the SwapTransformer at the inference time.

A notable challenge arises in overtaking as there is no precise metric for directly comparing different models. This challenge emerges from the inherent variability in decision-making within similar scenarios, wherein models may exhibit slight variations in their lane change decisions over a few sequential steps. Even minor divergences in decision-making (e.g., deciding on a right lane change instead of keeping the current lane) can result in substantial differences in the environmental states of different models, hence comparing the models to the rule-based driver on a step-wise basis does not accurately reflect the performance of models.

## 5  CONCLUSION

In this research paper, we tackled the problem of lane changing and overtaking on highways. We have introduced an AI approach using imitation learning called SwapTransformer. To train this model, 9 million frames containing overtakes with different traffic densities are collected in the simulation. This data is released as the OSHA dataset which is currently accessible by the public. SwapTransformer leverages dimension transposition, temporal and feature-based attention mechanisms, and auxiliary tasks like predicting future positions and distance matrices. In performance evaluations against MLP and transformer baseline models, SwapTransformer consistently outperformed them across diverse traffic densities, as evidenced by lap completion time, speed deviation from the limit, and overtaking score metrics, showcasing its robust generalization capabilities.

### REPRODUCIBILITY

We are committed to promoting transparency and reproducibility in our research. To facilitate the replication of our results, we provide comprehensive information about the code, data, and methods used in this study. Appendix provides more information about OSHA dataset and also the source

code used for data processing, model training, and evaluation is available in the supplementary materials of this paper.

## AUTHOR CONTRIBUTIONS

Will be available after the review phase.

## ACKNOWLEDGMENTS

Acknowledgments will be available after the review phase.

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

# A APPENDIX

## A.1 TRAVEL ASSIST CONTROLLER

Figure 5 shows different states and phases of the Travel Assist controller. Controller starts in the *None* state and upon receiving the lane change action, it goes to the *instantiated* state. In this state, the ego car starts signalling for a specific amount of time. Next, it goes to *Ready-to-change* state which the travel assist checks for safety. If it is safe to change, the ego car will start changing lane (*start the lateral movement*), otherwise, it will go to *Interrupt* followed by *Fail* states and then back to the *None* state. In simulation, Sumo has another sanity check for the lane change; while this does not exist in reality. In the last condition, Travel Assist controller checks if lane change occurred or not and based on that states of *Success* or *Fail* are defined.

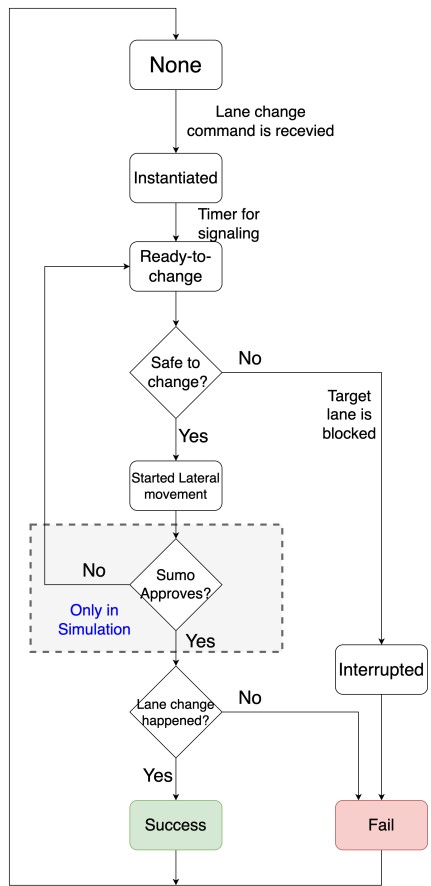

Figure 5: Travel Assist flowchart including different states.

## A.2 SIMULATION

Figure 6 illustrates the graphical user interface (GUI) of SimPilot in a highway scene. SimPilot communicates with the Python interface with a frequency of 50 Hz which is equal to 20ms in simulation timestep. The simulation environment provides object list data for 20 vehicles around the ego within the vicinity of 100m.

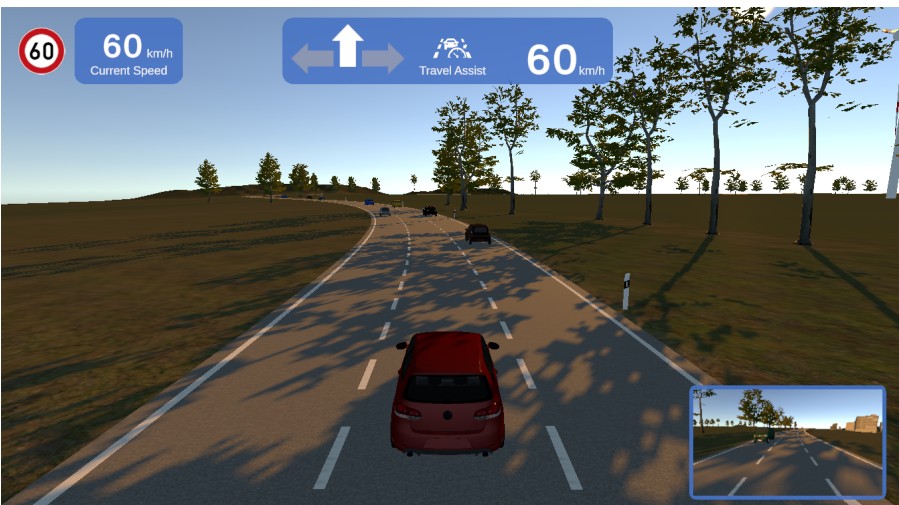

Figure 6: GUI of SimPilot including ego and other vehicles driving on a highway scene.

## A.3 RULE-BASED ALGORITHM

The rule-based driver is a simple physical model based on a set of rules and numerical thresholds that are determined by experiments and previous work on physical models controlling the vehicle Xu et al. (2022) Xu et al. (2022) Li et al. (2021). As explained in the paper, the rule-based algorithm has different components for decision-making. The precise definition of each component for a lane change can be described as:

- Safety: This includes the time to collision with the vehicle in front of ego in the current lane, the vehicle in front of the destination lane, and behind ego in both lanes. Plus, the distance between the ego and vehicles in front and behind it in both lanes should be greater than or equal to the safe distances defined.

- Speed gain: The purpose of a lane change in our approach is to drive faster and match the speed limit. Given all the available information about surrounding vehicles, estimate the speed at which ego will move after the lane change happens and compare it with the estimated speed of staying in the current lane. This condition is met if the ego can drive faster after a lane change. Or if there is an available right lane where ego can drive as fast in it, a right lane change will happen.

- Availability: Not only should there be a lane available next to the ego for a lane change to happen, but there should also be enough space for the ego to move to that lane. In other words, another vehicle should not be parallel to ego in the destination lane.

- Controller acceptance: On top of all the previous three conditions required for a reasonable lane change, our controller might reject a lane change command sent due to a sudden lane change or acceleration of surrounding vehicles.

The flowchart in Figure 7 represents the general policy of the rule-based driver. An overtake is defined as two lane changes, where ego will make a lane change, pass the vehicle in front, and then if there is enough space to get back to the original lane, another lane change will occur. As previously mentioned, the controller has safety features to avoid collisions. However, the safety of each lane change is also checked by considering the current state of the available lane. This improves the quality of lane changes with a higher success rate.

Figure 8 shows the input of the proposed rule-based algorithm. The six vehicles in the ego's current lane and (possibly) adjacent left and right lanes will be used to make a decision. For each vehicle we use the $\langle v_k, x_k, y_k, L_{ID_k}, m_k \rangle$ where $v_k \in \mathbb{R}^+ \cup \{0\}$ is the velocity for vehicle $k$. $x_k, y_k \in \mathbb{R}$ are the positions for vehicle $k$ defined in a local coordinate where the ego is at the origin $(0, 0)$. The length of $k^{th}$ vehicle is defined as $m_k$ where $m_k \in \mathbb{R}, \forall k \in [1, 20]$. The relative position of each vehicle can be found easily and we can label them as current-front, current-rear, left-front, left-rear,

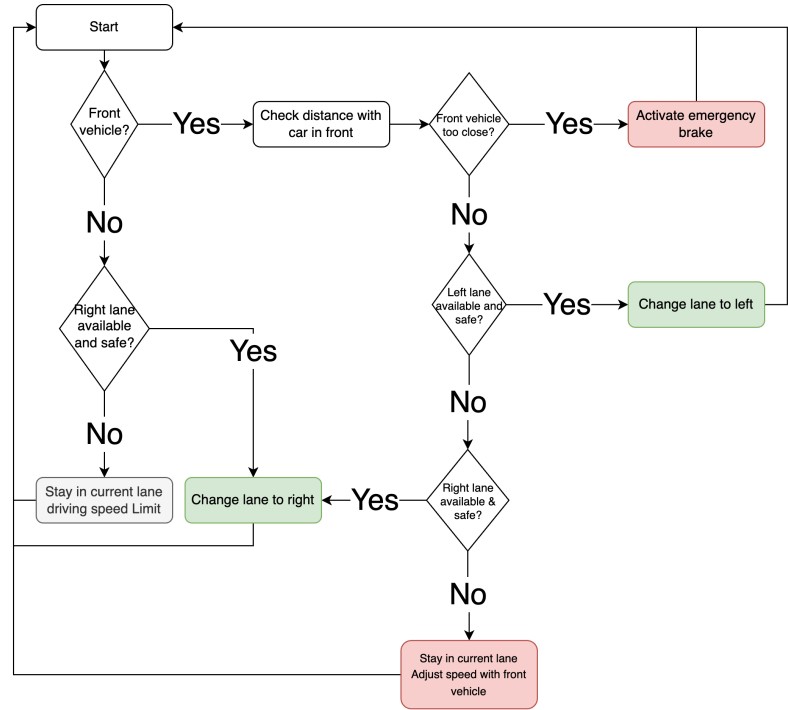

Figure 7: Rule-based algorithm flowchart

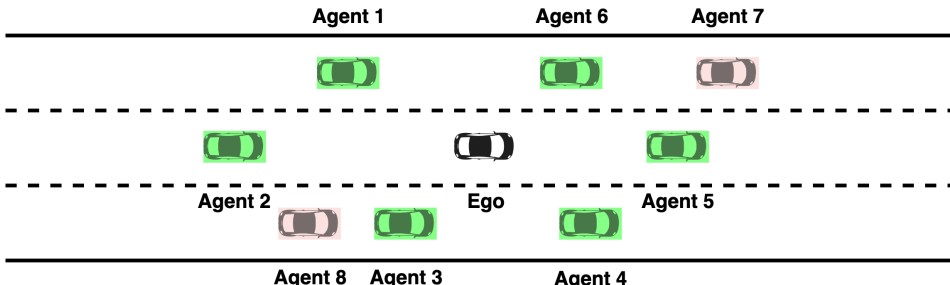

Figure 8: Rule-based algorithm observation

right-front, and right-rear. For simplicity, other agents available in the field of view of ego are not considered. In Figure 8, agents 7 and 8 are not observed by the rule-based driver.

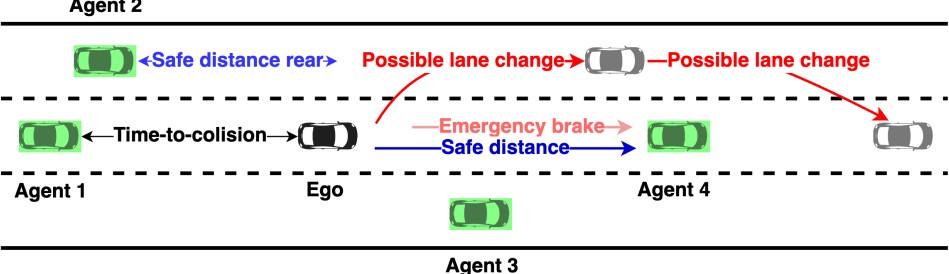

Figure 9: Rule-based safety metrics

To have a reasonable lane change, we also consider the speed of the vehicles in the destination lane. This also includes the distance to vehicles in the rear and front and the time distance (also known as

time of collision). Also, Figure 9 shows how an overtake will consist of two lane changes in a short period, as well as showing the emergency brake distance and safe distance that will be kept with the vehicle in front at all times. The safe distance in the rear is part of our additional requirement for a lane change. Even though the Travel Assist controller has internal safety conditions to perform a lane change we wanted to make sure that model learns the best behaviors possible.

It is worth noting that the rule-based algorithm was implemented based on many fine-tuning and tweaks on many parameters to cover a simple highway scenario. However, it needs lots of effort to have a rule-based approach for all different scenarios. The rule-based algorithm has limited access only to the vehicles adjacent to the ego (front, rear, left, and right) whereas the SwapTransformer has access to all the vehicles visible. The main reason for the rule-based approach is to have an automated way to generate ground truth data. As pointed out in Section A.7 (future works), the rule-based will be replaced by human drivers to collect realist data.

### A.4 DATASET

#### A.4.1 FUTURE POSITION

As explained in the paper, those future points are needed as ground truth for the auxiliary task purpose. Future position processing and transformation from global to local coordinate system are shown in:

$$(\boldsymbol{x}_{t+i},\ \boldsymbol{y}_{t+i})_{\text{local}} = \begin{bmatrix} \cos(\phi_t) & \sin(\phi_t) \\ -\sin(\phi_t) & \cos(\phi_t) \end{bmatrix} \begin{bmatrix} x_{t+i} - x_t \\ y_{t+i} - y_t \end{bmatrix}_{\text{global}} \qquad \forall 1 \le i \le 5, \qquad (8)$$

where $\boldsymbol{x}_{t+i}$ and $\boldsymbol{y}_{t+i}$ are the future processed local positions for $t^{th}$ sample in the dataset in the vector format. In this study, five future locations are considered for prediction. Hence the vector has five future positions such as $(\boldsymbol{x}_{t+i}$ and $\boldsymbol{y}_{t+i}) = \{(x_{t+1}, y_{t+1}), (x_{t+2}, y_{t+2}), \ldots, (x_{t+5}, y_{t+5})\}$. $\phi_t$ is the orientation for the ego car at time stamp $t$ in the global coordinate system. The second matrix is used to calculate the relative distance between the future and current location of the ego in the global coordinate system.

#### A.4.2 DATA PRUNING

During the pre-processing phase, some samples that result in collisions are removed from the dataset. This means if a collision happens between the ego and other vehicles at any time, the pre-processing phase trims that episode to avoid having those samples in the training dataset. In general, a collision may happen when other vehicles hit the ego vehicle, due to a sudden lane change. In addition, since the model is predicting five future points which are 2.5 seconds in the future, the end of all episodes is cut to have meaningful data for training. This results in 8,206,442 samples of data after pre-processing out of 8,970,692 raw samples. Also, a few new features are added to the raw dataset for the sake of training. These new features are future positions, future velocity, future lane change commands, and a matrix of distances between each pair of vehicles including the ego and others.

Figure 10 illustrates a few samples of the dataset. The first row shows the GUI sample of the surrounding environment and the second row shows the road curvature based on different lane IDs mapped to the segmentation of the drivable area. The third row displays how other dynamic objects(vehicles) are rasterized to the lane ID segmentation image with respect to the ego vehicle location. The data collected and shared in this dataset is one channel with a dimension of 50 pixels in width by 100 pixels in height. The resolution for the image is set as 0.5 meters per pixel. The ego vehicle is located at a fixed location in each image. The ego is horizontally at the center and vertically, it has 10 pixels offset below the center. The image data are stored in PNG format and ego and other vehicles' features and information are stored in pickle data frame format.

Figure 11 shows this idea, how the pre-processing phase adds those new lane changes and the transition class to the dataset. Before doing any pre-processing, only a single left lane change is detected in Figure 11a; however, after adding the new class; *Transition*, and artificial lane changes, the dataset becomes more balanced. This is illustrated in Figure 11b.

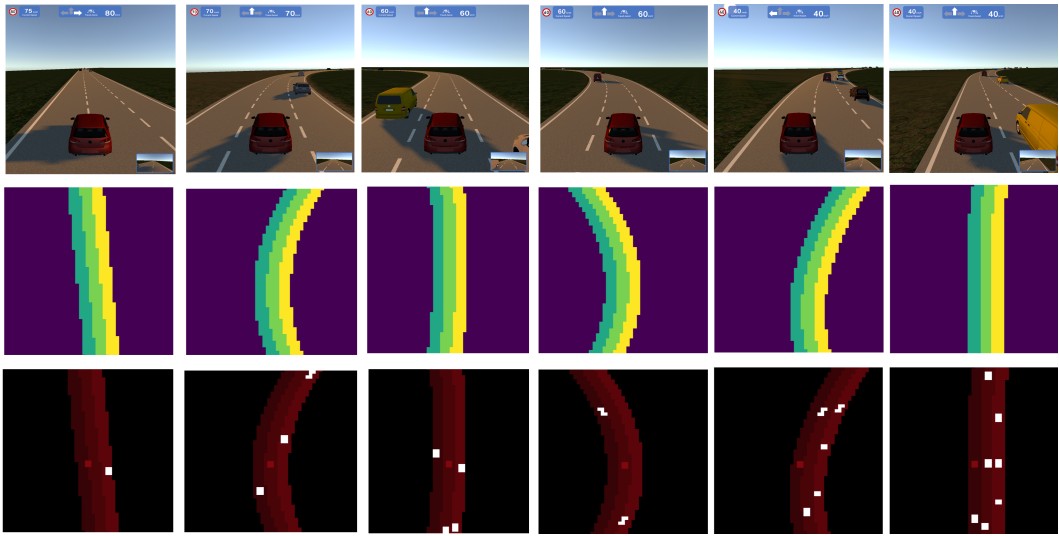

Figure 10: A few samples from the dataset AfterReview (2023).

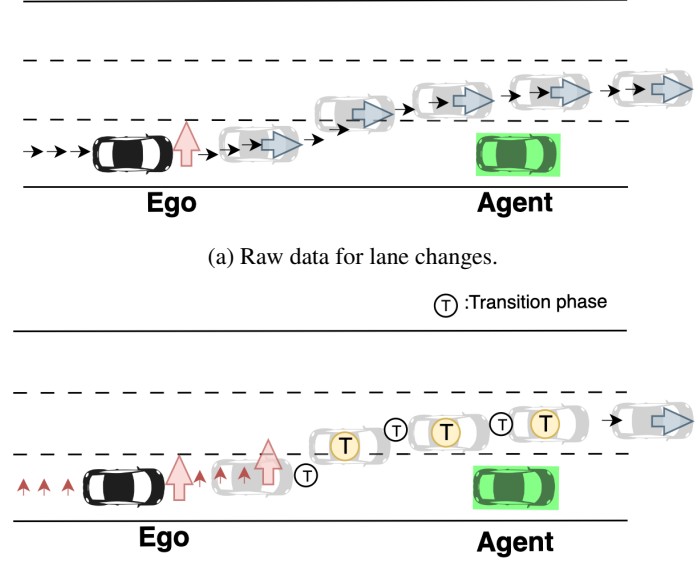

(a) Raw data for lane changes.

(b) Processed data for lane changes.

Figure 11: Difference between before pre-processing and after pre-processing on lane changes.

## A.5 MODEL DETAILS

The given algorithm, A.5 outlines a procedure for processing batches of data using a set of Transformer Encoders denoted as $\psi$. It operates on individual elements $X_i$ in a dataset batch, each of which is represented as a tuple $X_i = [t_i, \; d_i, \; f_i]$, where $t_i$ is the historical and temporal information, $d_i$ is an arbitrary embedded dimension with respect to data, and $f_i$ is the feature inputs with respect to ego vehicle and object list. The algorithm begins by applying Positional Encoding (PE) to the input $X_i$. Then, it iteratively applies the Transformer Encoders from the set $\psi$ to the encoded input $X_{i,\text{encoded}}$, with alternating transformations based on whether the index of the encoder is even or odd.

---

**ALGORITHM 1:** SwapTransformer

---

1  **for** *batch* in Dataset **do**
2     Let $X_i \in$ batch
3     $X_i = [t_i,\ d_i,\ f_i]$;
4     Let $\psi$ be a set of Transformer Encoders.
5     $X_{\text{i, encoded}} = PE(X_i)$
6     **for** *n in [ 1,2,... $|\psi|$]* **do**
7         **if** *n mod 2 == 0* **then**
8             H = $\psi_n(H^T)$;
9         **else**
10            H = $\psi_n(H)$;
11         **end**
12     **end**
13 **end**

---

### A.6   TRAINING AND EVALUATION

In more detail, Figures 12a and 12b demonstrate the map and road structure for training and inference scenes respectively. As illustrated, the inference scene completely has a different shape so we can better evaluate the generalization of the proposed model. Especially, the curvature of the evaluation scene was set to be different than the training scene while designing the evaluation scene.

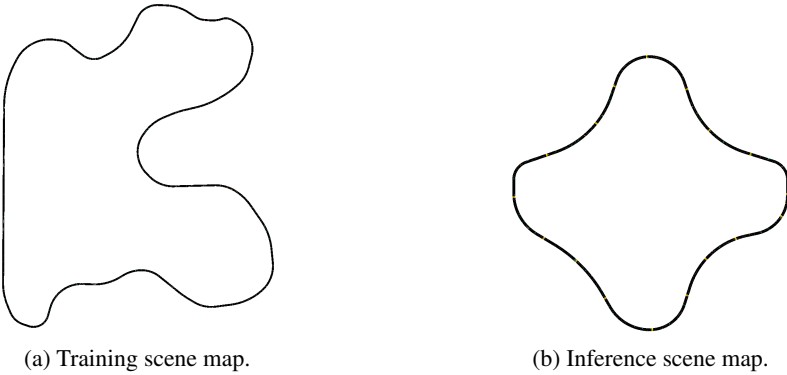

(a) Training scene map.           (b) Inference scene map.

Figure 12: Difference between training and inference scenes.

During the initial stages of model design and training, hyperparameter tuning was performed on the naive transformer without auxiliary loss or the swapping feature to optimize performance. Various versions of EfficientNet Tan & Le (2019) (b2 and b6) and ResNet (18, 50) were considered, along with a custom vision encoder. ResNet18 was chosen for its similar performance to other models with lower computational costs. Hyperparameter sweeps using WandB Biewald (2020) were conducted to determine optimal values for learning rate, scheduler, optimizer, batch size, activation layers, and regularization. Moreover, all models used the same seed at the training time for initialization to make sure the backbone was initialized with similar weights.

Figure 4 in section 4.3 demonstrated six samples in six different scenarios with different traffic behaviors. The prediction of future points is shown as yellow points as an auxiliary one. Velocity action is directly applied to the Travel Assist controller and the ego speed can be shown at the top left corner of each sample. The lane change command is also illustrated as signaling on the ego and as an arrow in the top layer. It can be observed that during lane changes, the behavior of future points is more aligned with the new decision to land on a new lane on the highway. Interestingly, one can see the future positions drifting towards another lane right before a lane change command is sent. This further proves the effect of auxiliary tasks on the performance of main tasks.

During the evaluation time, the same seed is used for all evaluations between different models. This brings fairness to the comparison. The randomness that applied at the inference and evaluation time accounted for traffic behavior like density, placement, and velocity of other agents. All these

randomnesses were defined based on some average and standard deviation (same mean and std for all of them in different situations).

## A.7 FUTURE WORKS

This is an ongoing project and based on the progress in the simulation platform next steps are split as:

- The first step is the implementation in reality and testing of the SwapTransformer on real-world highways. This phase of implementation in SimPilot is proved as a proof-of-concept. There already exists real-world data from highway scenarios collected by vehicles equipped with the Travel Assist module. However, a fleet of cars is already ready to collect more data for fine-tuning the SwapTransformer.

- Another future direction involves bringing the navigation information as an additional input to the SwapTransformer for entering and exiting highways. The aim of this task is to have more intelligent lane changing and overtaking considering the navigation commands.

- Motion forecasting of other dynamic vehicles plays a vital role in the planner module. Currently, a motion forecasting model is trained and available as an independent AI model to predict the future trajectories and positions of other vehicles on highways. We believe that feeding these predicted outputs as an additional input to the SwapTransformer will help the tactical planner to make wiser and safer lane change and overtaking decisions.

