# OpenReview forum: "SwapTransformer: Highway Overtaking Tactical Planner Model via Imitation Learning on OSHA Dataset"
_ICLR.cc/2024/Conference — Submitted to ICLR 2024_

### Official Review · Reviewer_kokN · 2023-10-28

**Soundness:** 2 fair
**Presentation:** 2 fair
**Contribution:** 2 fair
**Rating:** 5
**Confidence:** 3

**Summary:**

This paper proposes an ML planner approach to have the ego overtake other cars on highways.

The authors first collected a dataset through simulation. They used the SimPilot simulator and implemented a rule-based algorithm to have the ego overtake other agents. They used the rule-based driver as the ground truth expert.

The authors then designed a transformer-based architecture to make lane-change decisions. The inputs of the model are lane ID segmentation images and agent information. The main task of the model is to predict lane-change decisions and lane-change velocities. They also proposed two auxiliary tasks. One is to predict the ego vehicle's future positions, and the other one is to predict a distance matrix between all agents.

The model uses ResNet-18 to encode the lane ID image and uses an attention-based module to process agent information.

The authors evaluated the proposed model in the simulation environment. They compared against the following baseline methods: MLP only, transformer only, transformer + auxiliary tasks, and SwapTransformer. The result shows that the proposed model has the best performance in terms of speed difference, time to finish, and overtake ratio.

**Strengths:**

* The paper presents a good ML model architecture to make lane-change decisions.

* The proposed method has better performance than the baselines.

* Both the data and code will be released.

**Weaknesses:**

* The baselines used in this paper are all variants of the author's own methods. There are no comparisons against other state-of-the-art methods.

* The authors generated the ground truth with a rule-based driver. Does it mean the proposed method can only do as well as the rule-based method? What are the advantages of the proposed method over the rule-based method?

**Questions:**

* The proposed method can only do as well as the rule-based method. What are the advantages of the proposed method over the rule-based method?

---

> ### Author Response · Authors · 2023-11-17
> **Responses to reviewer kokN**
>
> Dear reviewer, first we appreciate the time and effort you spent on reviewing this paper. We tried to answer all of your concerns and comments for our paper, hope these answers make the paper clearer from your perspective. While we answered the comments here, we modified the paper to have the edited content accordingly. We went through both the weakness points and questions you had and we are trying to submit responses for both parts as long as they are not identical.
>
> ## Weaknesses:
>
>
> **_1) The baselines used in this paper are all variants of the author's own methods. There are no comparisons against other state-of-the-art methods._**
>
> We appreciate the reviewer for this concern. We would like to mention that most of the SOTA methods are focused on motion forecasting using end-to-end methods. Also, our main contribution to this work is to propose an approach for imitation learning especially behavioral cloning rather than focusing on motion forecasting. Moreover, we are focusing on modular autonomous driving stacks where high-level decision-making is the interface between the model and the low-level controller. One of the benefits of using a modular architecture is that we can apply it to real vehicles equipped with the Travel Assist controller. These make our approach and dataset unique compared to other works.
>
> **_2) The authors generated the ground truth with a rule-based driver. Does it mean the proposed method can only do as well as the rule-based method? What are the advantages of the proposed method over the rule-based method?_**
>
>
> We like to point out this concern is addressed in the questions section.
>
> ## Questions:
>
>
> **_1) The proposed method can only do as well as the rule-based method. What are the advantages of the proposed method over the rule-based method?_**
>
>
> The rule-based method serves as a surrogate for a human driver in our problem context to help us prove the Swap Transformer concept. However, the rule-based driver, even in the simplest scenarios, requires a  significant effort to develop and can't be extended to more intricate states (city traffic, entering and exiting highways, etc.). During simulation, the rule-base played an instrumental part in the data collection process, thus enabling our model to undergo training. In real-world scenarios, the rule-based driver is replaced by an expert driver.

---

> > ### Comment · Reviewer_kokN · 2023-11-19
> > **Thank you for your response**
> >
> > Thank you for your response. I will keep my score.

---

### Official Review · Reviewer_Qxdb · 2023-10-31

**Soundness:** 2 fair
**Presentation:** 2 fair
**Contribution:** 2 fair
**Rating:** 5
**Confidence:** 3

**Summary:**

This paper imitation learning as a planning approach for highway autonomy, focusing specifically on overtake and lane change maneuvers.

The first contribution is a custom synthetic dataset for the task, generated in a Unity-based simulation environment. The second contribution is the formulation of the problem: instead of tackling this problem as a motion planning or trajectory forecasting challenge, it is assumed that a controller exists for ego which can perform lane keeping, adaptive cruise control, and controlled lane changes, and that the   task is to provide the controller with inputs indicating when to perform lane changes and in which direction. The final contribution is a proposed model along with comparisons to a variety of baselines.

The dataset abstracts away the perception system, collecting lane segmentations and object lists from the environment. These inputs are passed through a CNN and transformer architecture with a few auxiliary tasks. The auxiliary tasks include ego trajectory prediction (represented as Bezier curve conrol points) and predicting the pairwise distances between all vehicles in the scene. The core task is predicting the lane change maneuver as well as future ego velocity five steps ahead.

The paper compares a variety of models. The proposed model, SwapTransformer, uses decomposed attention with transpose operations in between. The input object list is represented with two spatial dimensions, one for the input feature and one for time; alternating transformer encoders with self attention on either dimension resembles common approaches for handling 2D inputs (somewhat similar to e.g. Axial Attention, Salimans 2019, for example). The baselines are a transformer without auxiliary losses, a transformer with axis swapping, and a simple MLP baseline; the proposed model outperforms these baselines on a few metrics such as the speed prediction accuracy.

**Strengths:**

The dataset and formulation of the problem are fairly original in my experience. The complex problem of highway driving is broken down into a lower level controller ("Traffic Assist") with a few key capabilities and a decision-making layer, with the paper focused on the decision-making component. The dataset is generated in simulation specifically for this formulation of the problem. While less ambitious than approaches that tackle full imitation learning for the controller, this formulation is potentially simpler to deploy in a real-world system building upon existing driving assist capabilities.

Overall, the quality of the writing is high and the descriptions are clear, with the caveat that system rests on the Traffic Assist system, the implementation of which is not documented besides a high-level explanation of its inputs and outputs

**Weaknesses:**

The biggest weakness of this paper is the challenge of placing its results in a wider research context. While the description is clear and the technical work is sound, it reads like a technical report; by using a custom dataset with a custom problem and a custom model, it is impossible to compare this work to open source or publicly available baselines for the same problem.

At the same time, this paper cannot be described as a significant dataset contribution: the dataset is generated with complex rule-based and simulation setups, which may be challenging to reproduce and have limited applicability to other challenges or datasets.

Overall, the key change I would make to this paper would be to find a way to connect it to other publicly available systems or research. If the key contribution is meant to be the dataset, I would recommend evaluating other future forecasting and prediction models (e.g. Wayformer) on the dataset; if the key is the model, I would consider focusing on public future forecasting benchmarks. If the main contribution is meant to be the problem decomposition, I would compare an end-to-end trajectory forecasting system with this proposed Traffic Assist-based decomposed model. Without these sort of connections to existing literature, it will be challenging for the community to derive value from this work.

**Questions:**

One question I would like to understand in more detail is if there is prior work on the Traffic Assist system in this paper, and if so, if there are any public benchmarks that could be added to this paper.

Another question is whether this system has been tested outside of simulation or deployed to real world systems. If so, reports on the results of this work would strengthen the work significantly.

---

> ### Author Response · Authors · 2023-11-17
> **Responses to reviewer Qxdb**
>
> Dear reviewer, first we appreciate the time and effort you spent on reviewing this paper. We tried to answer all of your concerns and comments for our paper, hope these answers make the paper clearer from your perspective. While we answered the comments here, we modified the paper to have the edited content accordingly. We went through both the weakness points and questions you had and we are trying to submit responses for both parts as long as they are not identical.
>
> ## Weaknesses:
>
> **_1) The biggest weakness of this paper is the challenge of placing its results in a wider research context. While the description is clear and the technical work is sound, it reads like a technical report; by using a custom dataset with a custom problem and a custom model, it is impossible to compare this work to open source or publicly available baselines for the same problem._**
>
>
> We appreciate the reviewers for this concern. We would like to mention that this work is one of the first papers regarding highway lane changes with a close-to-reality simulated data. While we tried to look for other public available highway datasets to compare our work, we realized that most of the previous works are in the domain of motion forecasting such as WOMD and waymax. It's crucial to distinguish between motion forecasting and our work in this paper. Motion forecasting involves fixed predicted trajectories compared to ground truth, whereas our approach entails generating actions fed into a controller that dynamically creates motion and controls the vehicle.
>
> WOMD:
> Chen, K., Ge, R., Qiu, H., Ai-Rfou, R., Qi, C.R., Zhou, X., Yang, Z., Ettinger, S., Sun, P., Leng, Z. and Mustafa, M., 2023. WOMD-LiDAR: Raw Sensor Dataset Benchmark for Motion Forecasting. arXiv preprint arXiv:2304.03834.
>
> Waymax:
> Gulino, C., Fu, J., Luo, W., Tucker, G., Bronstein, E., Lu, Y., Harb, J., Pan, X., Wang, Y., Chen, X. and Co-Reyes, J.D., 2023. Waymax: An Accelerated, Data-Driven Simulator for Large-Scale Autonomous Driving Research. arXiv preprint arXiv:2310.08710.
>
>
>
> **_2) At the same time, this paper cannot be described as a significant dataset contribution: the dataset is generated with complex rule-based and simulation setups, which may be challenging to reproduce and have limited applicability to other challenges or datasets._**
>
> What sets our dataset apart is that the controller is adapted based on real-world scenarios, distinguishing it from other datasets or simulations. This dataset is designed for those focusing on high-level decision-making, where actions are interpreted through a motion controller. The dataset can also be applied as a motion forecasting problem. Furthermore, our focus is solely on high-level object lists from a top-down perspective. Unlike other motion forecasting datasets from Waymo, Lyft, and Argoverse that primarily concentrate on urban scenarios, ours offers a unique advantage by providing both high-level and low-level actions in diverse highway scenarios.
>
> Waymo:
>
> Ettinger, S., Cheng, S., Caine, B., Liu, C., Zhao, H., Pradhan, S., Chai, Y., Sapp, B., Qi, C.R., Zhou, Y. and Yang, Z., 2021. Large scale interactive motion forecasting for autonomous driving: The waymo open motion dataset. In Proceedings of the IEEE/CVF International Conference on Computer Vision (pp. 9710-9719).
>
> Argoverse:
>
> Wilson, B., Qi, W., Agarwal, T., Lambert, J., Singh, J., Khandelwal, S., Pan, B., Kumar, R., Hartnett, A., Pontes, J.K. and Ramanan, D., 2023. Argoverse 2: Next generation datasets for self-driving perception and forecasting. arXiv preprint arXiv:2301.00493.
>
> Lyft:
>
> Houston, J., Zuidhof, G., Bergamini, L., Ye, Y., Chen, L., Jain, A., Omari, S., Iglovikov, V. and Ondruska, P., 2021, October. One thousand and one hours: Self-driving motion prediction dataset. In Conference on Robot Learning (pp. 409-418). PMLR.

---

> ### Author Response · Authors · 2023-11-17
> **(Contd.) Responses to reviewer Qxdb**
>
> **_3) Overall, the key change I would make to this paper would be to find a way to connect it to other publicly available systems or research. If the key contribution is meant to be the dataset, I would recommend evaluating other future forecasting and prediction models (e.g. Wayformer) on the dataset; if the key is the model, I would consider focusing on public future forecasting benchmarks. If the main contribution is meant to be the problem decomposition, I would compare an end-to-end trajectory forecasting system with this proposed Traffic Assist-based decomposed model. Without these sort of connections to existing literature, it will be challenging for the community to derive value from this work._**
>
> **_While our dataset provides both high-level (lane change commands and speed) and low-level (trajectories), our main focus is on modular approaches. Unlike Wayformer or Multipath++ which are widely used for motion forecasting, we are tackling high-level decision-making for the real-world product (vehicle). The dataset is not the main contribution of this paper but it adds value to the community by providing a novel dataset that has both high-level and low-level actions. We agree with you and would like to compare our proposed approach with an end-to-end approach but this is out of the scope of this paper since we are targeting a specific controller with a high-level interface. We consider this evaluation as future work with another vehicle using a low-level trajectory interface.
>
> Wayformer:
>
> Nayakanti, N., Al-Rfou, R., Zhou, A., Goel, K., Refaat, K.S. and Sapp, B., 2023, May. Wayformer: Motion forecasting via simple & efficient attention networks. In 2023 IEEE International Conference on Robotics and Automation (ICRA) (pp. 2980-2987). IEEE.
>
> Multipath++
>
> Varadarajan, B., Hefny, A., Srivastava, A., Refaat, K.S., Nayakanti, N., Cornman, A., Chen, K., Douillard, B., Lam, C.P., Anguelov, D. and Sapp, B., 2022, May. Multipath++: Efficient information fusion and trajectory aggregation for behavior prediction. In 2022 International Conference on Robotics and Automation (ICRA) (pp. 7814-7821). IEEE._**
>
>
> ## Questions:
>
> **_1) One question I would like to understand in more detail is if there is prior work on the Traffic Assist system in this paper, and if so, if there are any public benchmarks that could be added to this paper._**
>
> To the best of our knowledge and based on the searches conducted, there doesn't appear to be a benchmark available for comparing models specifically in high-level decision-making processes, such as those addressed in the Travel Assist system discussed in the paper below:
>
> Paula, D., König, T., Bauder, M., Petermeier, F., Kubjatko, T. and Schweiger, H.G., 2022. Performance Tests of the Tesla Autopilot and VW Travel Assist on a Rural Road. In Transport Means 2022: Proceedings of the 26th International Scientific Conference-Part II (pp. 498-508). Kaunas University of Technology.
>
> We added this paper as a citation in the Travel-Assist section A.1 and 2.1. However, this paper addresses a different problem and system model.
>
>
>
> **_2) Another question is whether this system has been tested outside of simulation or deployed to real world systems. If so, reports on the results of this work would strengthen the work significantly._**
>
> Regarding the second question, our current focus involves advancing to the next stage by deploying our model to real-world scenarios with a travel assist application. This step will provide us with valuable insights, and we anticipate reporting results on the system's performance in real-world settings in the near future. This phase of the research already mentioned in the future direction in Appendix A.7. At this moment, a fleet of vehicles is taking care of the real-world data collection for the real-world training.

---

### Official Review · Reviewer_8dui · 2023-11-01

**Soundness:** 2 fair
**Presentation:** 3 good
**Contribution:** 2 fair
**Rating:** 5
**Confidence:** 3

**Summary:**

The authors propose a novel time-series transformer architecture for the task of predicting lane change probabilities for overtaking exo vehicles. Their approach functions in the imitation learning context where a rule-based expert is used to generate training data of when the ego vehicle should switch lanes (section 2.2.2 and appendix A.3). They propose a swap operation transposing time and feature dimensions between encoder layers (section 3.1) and use auxiliary pose and vehicle distance metrics to improve training (section 3.2). The learner is integrated in a larger Travel Assist system (section 2.1) which controls the ego vehicle while their component only seems to deal with predicting lane change probabilities.

The authors evaluate their method in simulated highway traffic scenarios (section 4) where speed difference, time to finish and overtake ratio as used as metrics to compare against MLP and transformer baselines (Table 2).

The authors also describe the OSHA (Overtaking in Simulated HighwAys) dataset (section 2.3) which they plan to release.

**Strengths:**

- Novel architecture which appears to beat baselines (however I feel there is some approach issues, see weaknesses).
- The work is well written and clear.
- Included ablations are useful for analyzing the components of the proposed network.

**Weaknesses:**

- There is no variance or confidence interval on the main results in Table 2 over repeated trials or different network seeds. Since some of the improvements are quite small compared to baselines (1-2%), it would be useful to know if the improvement is repeatable / consistent.
- I have some concerns on the training procedure: Was any form of validation set or early stopping used to help prevent overfitting and tune hyperparameters? Were multiple seeds used? The manuscript solely states that all models were trained for 300 epochs.
- I assume the Ours method in Table 2 is Transformer + Aux + Swap? Quite often is seems that the addition of the auxiliary loss or swap operation actually decreases performance but for some reason when combined together (ours approach) performance improves beyond the original transformer architecture. Can some intuition be given for this behavior?
- There does not appear to be any intuition as to why the swap operation would improve performance. This is further confusing by the fact that the transformer + swap architecture seems to harm performance versus the transformer architecture alone (Table 2).
- I feel that the evaluation metrics (speed difference, time to finish, overtake ratio) do not holistically evaluate the method. Although the proposed method finishes faster and overtakes more often is it possible that some of these were risky maneuvers (I.E. network predicted overtake when it shouldn’t of)? I think it would have been useful to show the model’s test prediction accuracy versus the expert to alleviate these concerns (I.E. is it switching lanes when it should be).
- Were the 50 test/evaluation episodes consistent across all models for fairness? I assume each was randomized but I would hope that the random conditions were repeated for all models.

**Questions:**

Questions are in part copied from the weaknesses section:

- I have some concerns on the training procedure: Was any form of validation set or early stopping used to help prevent overfitting and tune hyperparameters? Were multiple seeds used? The manuscript solely states that all models were trained for 300 epochs.

- I assume the Ours method in Table 2 is Transformer + Aux + Swap? Quite often is seems that the addition of the auxiliary loss or swap operation actually decreases performance but for some reason when combined together (ours approach) performance improves beyond the original transformer architecture. Can some intuition be given for this behavior?

- I did not seem to find any intuition as to why the swap operation would improve performance. Can the authors clarify this point?

- Were the 50 test/evaluation episodes consistent across all models for fairness? I assume each was randomized but I would hope that the random conditions were repeated for all models.

- The description of the rule-based expert is somewhat ambiguous (appendix A.3) but can the authors clarify what advantages the learner has over this rule-based system? Does the rule-based system have additional ground-truth inputs that the learner does not possess during run-time or is the learner significantly faster?

---

> ### Author Response · Authors · 2023-11-17
> **Responses to reviewer 8dui**
>
> Dear reviewer, first we appreciate the time and effort you spent on reviewing this paper. We tried to answer all of your concerns and comments for our paper, hope these answers make the paper clearer from your perspective. While we answered the comments here, we modified the paper to have the edited content accordingly. We went through both the weakness points and questions you had and we are trying to submit responses for both parts as long as they are not identical.
>
> ## Weaknesses:
>
> **_1) There is no variance or confidence interval on the main results in Table 2 over repeated trials or different network seeds. Since some of the improvements are quite small compared to baselines (1-2%), it would be useful to know if the improvement is repeatable / consistent._**
>
> we thank you for this concern you brought up. We agree with you in this regard. With this respect, we added average and standard deviation to Table 2 to explicitly address that. As shown in the updated Table 2, the standard deviation is reported for speed difference and time-to-finish. Those confidence intervals for the SwapTransformer (ours) for both metrics are relatively in range compared to other evaluated models.
>
>
> **_2) I have some concerns on the training procedure: Was any form of validation set or early stopping used to help prevent overfitting and tune hyperparameters? Were multiple seeds used? The manuscript solely states that all models were trained for 300 epochs._**
>
> We appreciate the reviewer for this concern. We are trying to address each of the points separately. At the beginning of this research, we did offline validation and early stopping. However, offline validation does not necessarily show the performance of the model in all possible situations at the inference time. We observed that those models that have very good metrics in validation, do not perform well in a live interaction in simulation. Also, comparing an ongoing trained model with a validation set needs to match the prediction with the ground truth exactly at the same time-slot, this comparison with validation does not seem reasonable for planning approaches. However, for research on vehicle motion forecasting, validation sets are pretty useful for early stopping or having a good understanding of the training phase. We added this information to the paper to Section A.6. In fact, we tried different versions of training before going into evaluation and inference. We did hyperparameter tuning on different aspects. For instance, on the naive transformer model, we tried different versions of ResNet such as 18, 50, and 101 and we realized that there are not that many differences between different versions. We also tried a customized vision encoder instead of Resnet but the performance was not as good as Resnet 18. On WandB, we tried different sweeping mechanisms on different hyperparameters such as the learning rate of the optimizer and scheduler. In addition to that, we tried a multi-head approach vs a single-head approach. In a multi-head approach, the optimizer, loss function, and schedulers were considered independent. Based on the training metrics (loss values), the result was better compared to a single-head approach. However, the generalization on inference was better using a single-head approach. Moreover, we tried different numbers of layers and neurons for each single head of the model to find the right number of layers to have a good performance and also to keep the model light enough to be able to run the model in the vehicle. All of these hyperparameters tuning were applied on the naive transformer before applying the auxiliary tasks and swap feature. This information was not mentioned in the paper due to space limitations. We appreciate your concern, and we added this information to Section 4.2 (possibly appendix Section A.6). Regarding the multiple seed concern (if I understand it correctly), the seed was mentioned for the evaluation. The seed was used for the simulation evaluation and it was fixed between different models. This guarantees to have the same pattern of randomness when we compare different models together. If the seed is mentioned as a concern to make sure that different models are initialized with same random values at the beginning, that is correct.

---

> ### Author Response · Authors · 2023-11-17
> **(Contd.) Responses to reviewer 8dui**
>
> **_3) I assume the Ours method in Table 2 is Transformer + Aux + Swap? Quite often is seems that the addition of the auxiliary loss or swap operation actually decreases performance but for some reason when combined together (ours approach) performance improves beyond the original transformer architecture. Can some intuition be given for this behavior?_**
>
>
>
> That is correct, Our method means Transformer + Aux + Swap, we fixed that in the paper to explicitly mention it.
>
>
>
> **_4) There does not appear to be any intuition as to why the swap operation would improve performance. This is further confusing by the fact that the transformer + swap architecture seems to harm performance versus the transformer architecture alone (Table 2)._**
>
> We are trying to address both 3 and 4 here:
>
> The combination of auxiliary tasks and dimension-swapping exhibits synergistic effects, enhancing the model's performance by leveraging both strategies concurrently. However, when individually trained—either with Transformer+Aux or Transformer+Swap—both models exhibit a tendency to underfit within the 300-epoch computational budget. In the swap transformer, we 'swap' dimensions between the time horizon and features. If trained longer, Transformer+Aux or Transformer+Swap by themselves would outpace the Transformer in evaluation. Hence, the Swap mechanism and auxiliary tasks are complementary in terms of faster convergence during training because of the additional signals they provide for each other. We mentioned these details in Section~4.3.
>
>
> **_5) I feel that the evaluation metrics (speed difference, time to finish, overtake ratio) do not holistically evaluate the method. Although the proposed method finishes faster and overtakes more often is it possible that some of these were risky maneuvers (I.E. network predicted overtake when it shouldn’t of)? I think it would have been useful to show the model’s test prediction accuracy versus the expert to alleviate these concerns (I.E. is it switching lanes when it should be)._**
>
> The concern raised regarding the evaluation metrics (speed difference, time to finish, overtake ratio) is valid. While our proposed method demonstrates faster completion and more frequent overtakes, there is a legitimate question about the potential inclusion of risky maneuvers, particularly if the network predicted overtakes when it shouldn't have. In our simulations with the Travel Assist controller, safety measures inherently prevent such risky actions, such as abrupt lane changes. Addressing your suggestion about comparing our model's predictions to a rule-based approach, it's important to note that the rule-based method introduces a level of randomness in behavior, making it challenging to precisely evaluate against our approach. The rule-based driver incorporates diverse behaviors, including maintaining safe distances within a lane, during lane changes. While we recognize the value of comparing our model's predictions to a rule-based system, the inherent variability in rule-based behavior complicates a straightforward assessment. We opted for the chosen evaluation metrics due to this complexity.
>
>
> **_6) Were the 50 test/evaluation episodes consistent across all models for fairness? I assume each was randomized but I would hope that the random conditions were repeated for all models._**
>
> We appreciate your valid concern. The same seed is used for all evaluation between different models. As you mentioned, we tried to put evaluation fairly between models. The randomness that we could apply at the inference/eval time accounted for traffic behavior like the density of other cars, placement of other cars, and velocity behavior. All of these randomness were defined based on some average and standard deviation (same mean and std for all of them).

---

> ### Author Response · Authors · 2023-11-17
> **(Contd.) Responses to reviewer 8dui**
>
> ## Questions:
>
> **_The description of the rule-based expert is somewhat ambiguous (appendix A.3) but can the authors clarify what advantages the learner has over this rule-based system? Does the rule-based system have additional ground-truth inputs that the learner does not possess during run-time or is the learner significantly faster?_**
>
> We thank you for bringing up this concern. The rule-based is updated in the paper to make it clearer.
> Also, We would like to mention that this research has two aspects and proof of concepts; first, proving the concept of the swaptransformer in simulation, and second implementing the swaptransformer in reality on the real-world car. As we are addressing comments now, there is a fleet of vehicles performing data collection for the second phase of this research. The rule-based algorithm that was used in the first phase was implemented based on many fine-tuning and tweaks on many parameters to just cover a simple highway scenario. However, it is not possible to have a rule-based for all different scenarios. The rule-based algorithm has limited access only to the vehicles adjacent to the ego (front, rear, left, and right) whereas the SwapTransformer has access to all the vehicles visible. Regarding the second phase of this project, since it needs real-world data, a rule-based driver is not applicable and a human driver is needed to collect all those ground truth. Regarding the first part, at the very beginning of the research, a human driver was used to collect ground truth data behind simulation systems; however, this task was very overwhelming. As a result, lots of effort was spent to design a rule-based expert only for this highway scenario to have an automated way to generate ground truth data. The rule-based driver, even in its current form, is very sensitive to parameters, and tuning these parameters and defining them requires a lot of effort. We believe that extending this approach to more complicated scenarios is difficult due to cost of development and fine-tuning. We may use the rule-based driver on a highway scene or if you have another scenario like roundabout, you may use rule-based on that one as well. However, you may need a much more complex rule-based to handle both highway and roundabout scenarios, while it is possible to train a single model on all data to handle both cases.

---

> > ### Comment · Reviewer_8dui · 2023-11-20
> >
> > 1) The results for the __MLP__ baseline vs. __Ours__ look somewhat okay, but the __Transformer only__ vs. __Ours__ have relatively large standard deviations for their difference in means and so I would be less confident on these models consistently outperforming the transformer baseline. Is this variance calculated over the 50 evaluation episodes or repeated trials of the same 50 episodes (though ideally with different network seeds)? For the former case, I could see the difference being large because I assume episodes have different degrees of difficulty assuming vehicles and the scene are randomized. However, if these results are for model repetitions of the evaluation episodes, they seem excessively large to be confident that the proposed method outperforms the transformer baseline consistently.
> >
> > To clarify: When I referred to using “different network seeds”, I was asking if the network was re-trained and tested with different seeds to calculate these variance results. The wording in the manuscript text seems to indicate that only 1 seed was used though perhaps I am misinterpreting it.
> >
> > 2) If a static validation set is not suitable, I could see using validation trajectories in the interactive simulation environment for tuning before final evaluation. Thank you for describing what parameters were tuned, however can you clarify if the tuning was done on separate validation trajectories (whether offline or in interactive simulation) or was tuning done directly on the final test trajectories?
> >
> > 3 + 4) Thank you for clarifying what is __Ours__ approach is composed of and giving some intuition as to why the adjustments perform individually worse.
> >
> > 5 - 7) Thank you for addressing my concerns.
> >
> > Could you also please also answer this question which seems to be missing in your reply: __I did not seem to find any intuition as to why the swap operation would improve performance. Can the authors clarify this point?__

---

> > > ### Author Response · Authors · 2023-11-20
> > > **Responses to reviewer 8dui (second round)**
> > >
> > > Dear reviewer, we appreciate it once again for the prompt feedback you mentioned. We apologize we missed that question about the swapping feature intuition.
> > > Here is our response:
> > >
> > > ## Points:
> > >
> > >
> > > **_1) The results for the MLP baseline vs. Ours look somewhat okay, but the Transformer only vs. Ours have relatively large standard deviations for their difference in means and so I would be less confident on these models consistently outperforming the transformer baseline. Is this variance calculated over the 50 evaluation episodes or repeated trials of the same 50 episodes (though ideally with different network seeds)? For the former case, I could see the difference being large because I assume episodes have different degrees of difficulty assuming vehicles and the scene are randomized. However, if these results are for model repetitions of the evaluation episodes, they seem excessively large to be confident that the proposed method outperforms the transformer baseline consistently._**
> > >
> > >
> > > We provide visual justification through videos shared on GitHub to demonstrate better performance. Despite variations in the 50 evaluation episodes, generated with the same seed but not identical due to inherent randomness and interactive environment dynamics, the mean in our model is lower, and models are close in terms of standard deviation. These variations encompass the degree of freedom in initial placement and lane configurations too. Actions made by the ego vehicle introduce new states (or randomness), even if episodes are similar identical initially. We believe 50 episodes capture these variations, and qualitative evaluation supports our method's consistent outperformance, further detailed in the shared videos on the GitHub/paper website post-review.
> > >
> > >
> > > **_2) To clarify: When I referred to using “different network seeds”, I was asking if the network was re-trained and tested with different seeds to calculate these variance results. The wording in the manuscript text seems to indicate that only 1 seed was used though perhaps I am misinterpreting it._**
> > >
> > > You are right about it, regarding re-training or training between different models, only one seed is used at that phase.
> > >
> > >
> > > **_3) If a static validation set is not suitable, I could see using validation trajectories in the interactive simulation environment for tuning before final evaluation. Thank you for describing what parameters were tuned, however can you clarify if the tuning was done on separate validation trajectories (whether offline or in interactive simulation) or was tuning done directly on the final test trajectories?_**
> > >
> > > Thanks for bringing up this concern, your point is valid and in our current developing stack, we evaluate our models during training using online validation trajectories using the Sim. environment.  All hyperparameter tunings were done on the validation set during the training. There was no access to the test environment at the training/hyperparameter tuning phase.
> > >
> > >
> > > **_4) Could you also please also answer this question which seems to be missing in your reply: I did not seem to find any intuition as to why the swap operation would improve performance. Can the authors clarify this point?_**
> > >
> > > The concept behind the Swap Transformer, or the swapping operation, is to establish a correlation between historical context and object list information. The historical data extends back 5 seconds, encompassing details about vehicles and their surroundings in the object list. Our approach involves not only applying attention across the time horizon but also across features. By swapping between axes, we aim to associate information between time and features. This dimension-swapping diversifies the network attention, leading to improved overall performance.

---

> > > > ### Comment · Reviewer_8dui · 2023-11-20
> > > >
> > > > Thank you for replying to my feedback again.
> > > >
> > > > I have raised my score slightly based on your reply though still feel that it is slightly below the bar for acceptance unfortunately. I still do not feel overly confident that the proposed method beats baselines with high probability given their close proximity in value and lack of variance reported over repeated test episode sets using different network seeds.

---

> > > > > ### Author Response · Authors · 2023-11-21
> > > > > **Thank you**
> > > > >
> > > > > We appreciate all the constructive feedback you shared with us. All points are valuable and we are sure that this will help the paper quality.

---

### Author Response · Authors · 2023-11-17
**Response to all reviewers**

Dear reviewers,
We would like to thank you for the time and effort you spent reviewing this paper by pointing to all aspects of that. We went through all the mentioned items in the review phase. All responses will be mentioned specifically to each reviewer. The manuscript has been updated based on those feedbacks, hence please consider the updated one. Most updated parts are mentioned in the concern and it is mentioned where it changed.

---

### Meta-Review · Area_Chair_KVmP · 2023-12-07

**Metareview:**

This paper considers learning lane change decisions. A rule-based controller generates demonstrations in a driving similator. A transformer-based imitation learning approach is developed to imitate these decisions. When trained to also provide predictions on auxiliary tasks (e.g., future point prediction), the approach provides better performance than multi-layer perceptron and transformer-only baselines. A main strengths of the paper are the construction a concrete high-level decision problem/dataset for autonomous driving, establishing some key benchmarks, and developing a novel approach that performs well. As weaknesses: (1) the value of the dataset/benchmark is limited by the fact that the demonstrations are from a rule-based controller rather than real drivers; (2) separately adding swap or auxilliary training to the Transformer model seems to degrade performance compared to the transformer alone---this is not well explained; and (3) there aren't comparisons to any more sophisticated SOTA imitation learning approaches on this task. These weaknesses outweight the strengths of the paper in its current form and prevent an acceptance recommendation.

**Justification For Why Not Higher Score:**

The synthetic nature of the dataset limits its standalone value and comparisons with (closer to) SOTA imitation approaches are needed.

**Justification For Why Not Lower Score:**

N/A

---

### Decision · Program_Chairs · 2024-01-16

Reject